# FreeAL: Towards Human-Free Active Learning in the Era of Large Language Models

**Ruixuan Xiao[1], Yiwen Dong[1], Junbo Zhao[1], Runze Wu[2]**
**Minmin Lin[2], Gang Chen[1], Haobo Wang[1]***
[1]Zhejiang University, Hangzhou, China
[2]NetEase Fuxi AI Lab, Hangzhou, China
{xiaoruixuan,dyw424,j.zhao,cg,wanghaobo}@zju.edu.cn
{wurunze1,linminmin01}@corp.netease.com

## Abstract

Collecting high-quality labeled data for model training is notoriously time-consuming and labor-intensive for various NLP tasks. While copious solutions, such as active learning for small language models (SLMs) and prevalent in-context learning in the era of large language models (LLMs), have been proposed and alleviate the labeling burden to some extent, their performances are still subject to human intervention. It is still underexplored how to reduce the annotation cost in the LLMs era. To bridge this, we revolutionize traditional active learning and propose an innovative collaborative learning framework FreeAL to interactively distill and filter the task-specific knowledge from LLMs. During collaborative training, an LLM serves as an active annotator inculcating its coarse-grained knowledge, while a downstream SLM is incurred as a student to filter out high-quality in-context samples to feedback LLM for the subsequent label refinery. Extensive experiments on eight benchmark datasets demonstrate that FreeAL largely enhances the zero-shot performances for both SLM and LLM without any human supervision. The code is available at https://github.com/Justherozen/FreeAL.

## 1 Introduction

Modern machine learning models typically require a huge collection of precisely labeled data, which can be a labor-intensive and time-consuming process. Even worse, it can be unrealistic in some practical scenarios that demand much expertise, such as medical diagnosis and industrial applications. To this end, a plethora of approaches have been investigated to reduce the burden of annotation, including semi-supervised learning (Sohn et al., 2020; Berthelot et al., 2019), learning with label noise (Han et al., 2018; Li et al., 2020), and so on. Amongst them, active learning (Ein-Dor et al.,

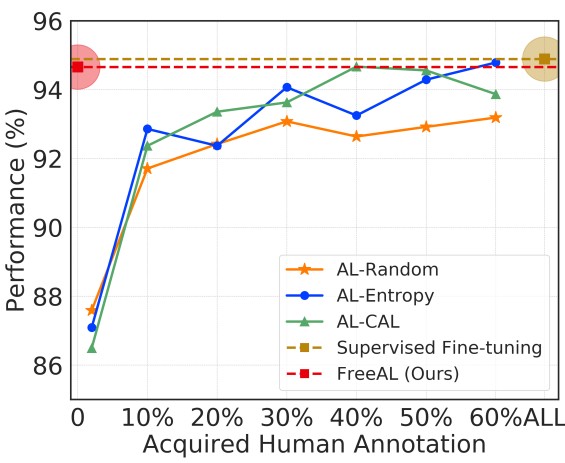

Figure 1: Comparisons of FreeAL with traditional active learning (AL) algorithms and supervised fine-tuning on the SST-2 dataset. FreeAL surpasses all the active learning rivals and achieves near-supervised performance without human annotation.

2020; Yuan et al., 2020; Margatina et al., 2021) is a prominent solution that interactively queries an external expert or oracle to mark the new data points that the model wants to learn from. These methods alleviate the labeling burden to some extent but still require human efforts in the annotation or construction of the oracle to start with.

The recent prevalent large language models (LLMs) (Ouyang et al., 2022; Thoppilan et al., 2022; OpenAI, 2023), such as ChatGPT and PaLM (Chowdhery et al., 2022), have exhibited strong zero-shot learning ability by proper prompt design, yet becoming a new remedy for data efficiency. Even more inspiringly, LLMs emerge with the so-called in-context learning (ICL) (Brown et al., 2020) ability to learn from a few task-related labeled samples for boosted performance. Despite the promise, some studies (Bang et al., 2023) find that LLMs tend to underperform compared to fine-tuned small language models (SLMs) on challenging tasks, which is also verified in our empirical

---
*Corresponding author.

studies (Table 3). One possible reason is that ICL can not fully exploit supervised training samples due to limited context length. Moreover, their extremely large size and limited accessibility also hinder their training and generalization on specific tasks. To date, it is still questionable how can we generalize to downstream tasks with the least human annotation in the era of LLMs.

In this work, we present a novel collaborative learning paradigm FreeAL that revolutionizes traditional active learning by interactively distilling and filtering the task-related knowledge from the LLMs. Our intuition is that, while LLMs are hard to fine-tune, they are competent zero-shot learners (Wei et al., 2022; Kojima et al., 2022) and can provide coarse-grained knowledge for downstream tasks. On the other hand, SLMs are effective weak learners (Li et al., 2020) that can distill valuable clean samples from noisy supervision. To integrate LLMs and SLMs synergistically as a whole, we design a collaborative training framework where LLM operates as an active annotator infusing its knowledge and the SLM acts as a student to filter out the high-quality input-label pairs to feed back the LLM for subsequent label refinery. Empirically, FreeAL iteratively boosts the unsupervised performance of both SLMs and LLMs during collaborative training for transductive and inductive settings. As depicted in Figure 1, FreeAL allows us to achieve an extraordinary annotation-performance trade-off by obtaining competitive results on par with the supervised counterparts while fully eliminating human annotation costs.

Overall, our main contributions can be summarized as follows,

- To the best of our knowledge, we are among the first to overhaul traditional active learning in the era of LLMs for boosted generalization performance *without any human supervision*.

- We propose a novel collaborative learning framework called FreeAL to employ the LLMs as active annotators and the SLMs as weak filters to interactively distill the task-related knowledge from the LLMs.

- Our proposed FreeAL largely improves the unsupervised learning performance for both the LLMs and the SLMs, even approaching the supervised counterparts in some scenarios. Our results prove the feasibility of human-free active labeling in the era of LLMs.

## 2 Related Work

### 2.1 Prompt-based Zero/Few-shot Learning

The emergent ability of LLMs has sparked heightened interest in prompt-based zero-shot and few-shot learning (Ye et al., 2021; Schick and Schütze, 2021). Instead of fine-tuning on massive downstream data, in-context learning (ICL) (Brown et al., 2020), which suits LLMs to new tasks with few-shot input-label exemplars as demonstrations without training, has shown promising few-shot performance. It has been further improved by later works (Liu et al., 2022; Lu et al., 2022; SU et al., 2023).

On the other hand, zero-shot learning is much more challenging without task-specific data. Direct steering LLMs for predictions without in-context demonstrations can lead to significantly degraded performance (Gao et al., 2021). To bridge this, some methods (Wei et al., 2022; Sanh et al., 2022; Xu et al., 2022) adopt instruction tuning with a multi-task paradigm to further pre-train the LLMs with a collection of different tasks in shared prompting templates. However, these methods require cumbersome training for LLMs and the overwhelming bulk of cross-task human annotations. Another new line of research (Ye et al., 2022a; Meng et al., 2022; Ye et al., 2022b) endeavors to ameliorate zero-shot learning merely via dataset generation, while the synthesized data commonly involves a notable portion of low-quality samples and misses the nuanced semantics present in the original data. In our work, we take inspiration from active learning with an innovative viewpoint to distill and filter the rich knowledge from LLMs for boosted zero-shot generalization performance.

### 2.2 Active Learning

Active learning (AL) is a prevailing paradigm in various NLP tasks (Yuan et al., 2020; Zhao et al., 2020; Shelmanov et al., 2021; Wang et al., 2022) that aims to reduce labeling effort by selecting only the most useful examples to annotate. In each iteration of active learning, a model is trained on the currently labeled data and then tasked with selecting the most informative yet-to-be-labeled data point to be labeled for boosted performance. Based on different querying strategies (Settles and Craven, 2008), existing traditional active learning methods can be categorized into uncertainty-based methods (Prabhu et al., 2019; Margatina et al., 2021) and diversity-based methods (Sener and Savarese, 2018; Ru et al., 2020; Ash et al., 2020). While these

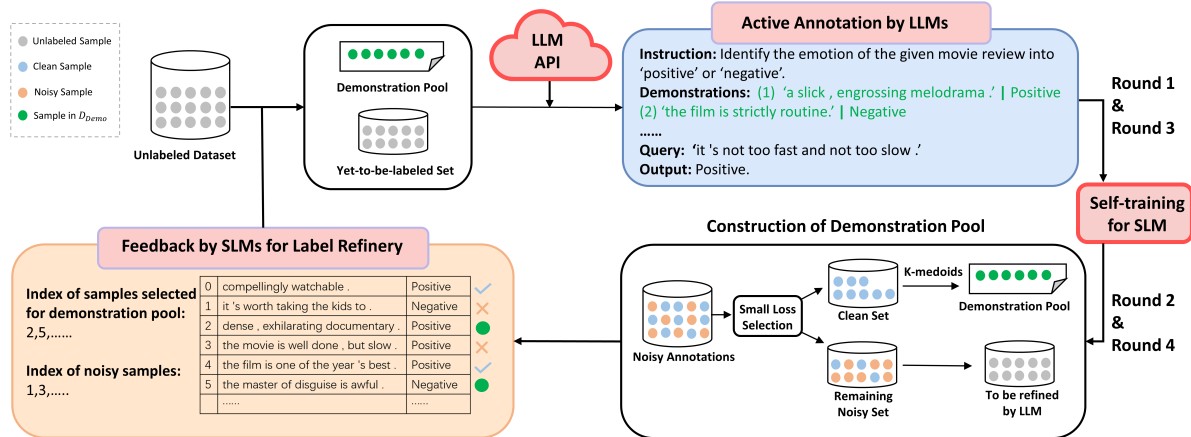

Figure 2: Overview of FreeAL. In each collaborative training loop, the LLM serves as an active annotator imbuing its knowledge. Then the SLM is employed as a filter to distill the task-related knowledge with robust self-training from LLM and filter out a high-quality demonstration pool $\mathcal{D}_{\text{demo}}$ to feedback the subsequent label refinery of LLM.

methods relieve the annotation burden to some extent, they still count on human experts as expensive supervision sources to start with. To overcome this high cost, we investigate the opportunities of leveraging the rich knowledge of LLMs as a low-cost supervision source for boosting generalization performance without human effort.

## 3 Background

We consider unsupervised classification tasks *without human annotations*. Given an unlabeled training dataset $\mathcal{D}_{\text{train}} = \{x_i\}_{i=1}^n$ with $n$ samples, where $x \in \mathcal{X}$ is the input text and the corresponding ground-truth label $y \in \mathcal{Y} = \{1, \ldots, C\}$ is inaccessible. Our task is to predict the true label for both the training dataset $\mathcal{D}_{\text{train}}$ and test dataset $\mathcal{D}_{\text{test}}$. Our framework employs a pre-trained large language model (LLM) $\mathcal{P}$ and a downstream small language model (SLM) $\mathcal{S}$. For the LLM, we define a natural language template $T(\cdot)$ which contains additional task-related information and a verbalizer $V(\cdot)$ which maps each class label in $\{1, \ldots, C\}$ to a pre-defined token in the prompt. For the fine-tuning of SLM $\mathcal{S}$ with parameters $\theta$, we adopt the cross entropy loss $l_i = -\sum_{j \in Y} \tilde{y}_i^j \log S^j(x_i, \theta)$ for training, where $S^j(x_i, \theta)$ is the $j$-th entry of SLM's output softmax probability for the input $x_i$ with the pseudo label $\tilde{y}_i^j$.

**Few-shot In-context Learning.** When supervised data are available, we can directly employ the few-shot ICL for inference. In concrete, given a demonstration supporting pool $\mathcal{D}_{\text{demo}} = \{x_i^{\text{demo}}, \tilde{y}_i^{\text{demo}}\}_{i=1}^m$ for prompt retrieval

during ICL, we construct a prompt including a test input $x^{\text{test}}$ and $m$-shot in-context examples $\{(x_j^{\text{demo}}, \tilde{y}_j^{\text{demo}})\}_{j=1}^m$ retrieved from $\mathcal{D}_{\text{demo}}$ as the demonstration. The final prompt steers the LLM and the prediction is obtained via,

$$\arg\max P_{y \in Y}(V(y) \mid T(x_1^{\text{demo}}, \tilde{y}_1^{\text{demo}}), \\ ..., T(x_m^{\text{demo}}, \tilde{y}_m^{\text{demo}}), T(x^{\text{test}})) \quad (1)$$

Despite the simplicity, the success of ICL largely hinges on the demonstration pool $\mathcal{D}_{\text{demo}}$, which requires human efforts of careful annotation for every individual scenario and can be particularly annoying for challenging tasks. To bridge this gap, we resort to our proposed plug-and-play method FreeAL without involving any human supervision.

## 4 FreeAL

In this section, we introduce our proposed framework FreeAL which investigates the opportunity for human-free active learning in the LLMs era. In contrast to traditional active learning that requests human annotation in each training loop, FreeAL employs LLMs as weak annotators. In each training loop, we alternate the following steps:

1. Active labeling of the to-be-labeled samples via LLMs based on the feedback from SLMs.

2. Training weakly supervised SLMs to distill the task-related knowledge from noisy annotations of LLMs and in turn feedback to them.

The overview of the FreeAL framework is displayed in Figure 2 and its overall pipeline is also shown in Algorithm 1. In what follows, we will elaborate on our FreeAL framework minutely.

## 4.1 Active Labeling by LLMs

In this step, we leverage the strong in-context learning ability of LLMs to assign weak labels to unsupervised training corpora. In particular, the core challenge lies in the construction of a proper prompt containing demonstration samples. To this end, we introduce two practical strategies for the different life cycles of FreeAL.

**Initial Annotation by Self-generated Demonstration.** At the initial round of FreeAL, we are given a purely unsupervised training set $\mathcal{D}_{\text{train}}$. To enable pseudo-labeling via LLMs, we may directly perform zero-shot ICL without access to a demonstration pool $\mathcal{D}_{\text{demo}}$. However, such a strategy can largely impede the knowledge-distilling process of SLMs due to shoddy initial annotations. To remedy this, we design a novel self-generated demonstration technique by virtual data generation. Notably, when given some unlabeled samples and task-descriptive instructions, humans can imitate the expression styles of these texts and leverage their own knowledge to generate similar label-aware samples. Motivated by this, we steer LLMs to first mimic the format of unlabeled samples from $\mathcal{D}_{\text{train}}$, which is important to ICL according to recent research (Min et al., 2022), and then generate new label-aware examples to construct the initial $\mathcal{D}_{\text{demo}}$.

Specifically, the data-generation prompt contains a hand-crafted task-descriptive instruction $\rho_{\text{gen}}$ that explains the task background and $Q$ randomly-selected unlabeled samples $c_{\text{gen}}$ from $\mathcal{D}_{\text{train}}$ as prototypes to imitate. An example of the prompt is shown in Appendix B.3. The generation process can be formulated as,

$$\{(x^{\text{gen}}, \tilde{y}^{\text{gen}})\} \leftarrow P(\rho_{\text{gen}}, T(c_{\text{gen}})) \qquad (2)$$

The generated samples constitute the generated dataset $\mathcal{D}_{\text{gen}} = \{(x^{\text{gen}}, \tilde{y}^{\text{gen}})\}$, which is then used as demonstration pool (i.e., $\mathcal{D}_{\text{demo}} = \mathcal{D}_{\text{gen}}$) for the subsequent labeling. Next, we follow the standard ICL pipelines with demonstration selection (Liu et al., 2022). Each prompt contains $m$-nearest-neighbors from $\mathcal{D}_{\text{demo}}$ with the highest embedding similarity to $x_i$. The ICL process follows Eq.(1). With the demonstrations seen in the prompt, the LLM is able to provide passable initial annotations $\tilde{y}$ of the training dataset $\mathcal{D}_{\text{train}} = \{x_i, \tilde{y}_i\}_{i=1}^{n}$, the annotations $\tilde{y}$ are employed as pseudo-labels for the subsequent training of SLM.

---

**Algorithm 1** Pipeline of FreeAL

**Input:** Unlabeled dataset $\mathcal{D}_{\text{train}} = \{x_i\}_{i=1}^{n}$; pre-trained LLM $\mathcal{P}$ and a downstream SLM $\mathcal{S}$;

1: $round \leftarrow 1$
2: **while** not convergent **do**
3:      # For LLM: active annotation
4:      **if** $round = 1$ **then**
5:          # Initial self-generated demonstration
6:          Generate $\{(x^{\text{gen}}, \tilde{y}^{\text{gen}})\}$ as Eq.(2);
7:          $\mathcal{D}_{\text{demo}} = \mathcal{D}_{\text{gen}} = \{(x^{\text{gen}}, \tilde{y}^{\text{gen}})\}$;
8:      **else**
9:          Receive $\mathcal{D}_{\text{demo}}$ from SLM;
10:      **end if**
11:      In-context learning as Eq.(1) for labeling;
12:      $round \leftarrow round + 1$;
13:      # For SLM: knowledge distillation
14:      Robust self-training as Eq.(3)
15:      # Construction of $\mathcal{D}_{\text{demo}}$
16:      Filter out class-wise clean subset $\mathcal{D}_{\text{clean}}^{j}$
17:      Adopt k-medoids on $\mathcal{D}_{\text{clean}}^{j}$ for $\mathcal{D}_{\text{demo}}^{j}$
18:      $\mathcal{D}_{\text{demo}} = \cup_{j \in Y} \mathcal{D}_{\text{demo}}^{j}$
19:      $\mathcal{D}_{\text{noisy}} = \mathcal{D}_{\text{train}} \setminus (\cup_{j \in Y} \mathcal{D}_{\text{clean}}^{j})$
20:      Feed $\mathcal{D}_{\text{demo}}$ and $\mathcal{D}_{\text{noisy}}$ back to LLM
21:      $round \leftarrow round + 1$;
22: **end while**

---

**Refined Annotation in Subsequent Rounds.** In the later rounds, the SLM $\mathcal{S}$ is trained using the weak annotation given by the LLM $\mathcal{P}$. Meanwhile, the SLM filters out a high-quality demonstration pool $\mathcal{D}_{\text{demo}}$ as feedback; details are shown in Section 4.2. Then with a high-quality $\mathcal{D}_{\text{demo}}$, the LLM $\mathcal{P}$ re-annotates the remaining noisy-prone samples via few-shot ICL according to Eq. (1).

## 4.2 Knowledge Distillation by SLMs

Given the acquired weak annotations from LLM, it is difficult for the LLM to distinguish its own errors due to the confirmation bias. Fortunately, previous studies (Han et al., 2018; Li et al., 2020) in weakly-supervised learning have shown that deep models have the potential of detecting noisy samples during the training procedure. Therefore, after receiving weak labels, our intention is two-fold: (i)-train a strong and robust downstream SLM that maximally distills task-specific knowledge; (ii)-employ the derived SLM to filter out a high-quality demonstration pool to feedback LLM.

### 4.2.1 Robust Self-training

Motivated by the memorization effect of DNNs (Zhang et al., 2017), the SLM tends to first fit easy patterns in the early stage of training. Thus, noisy samples mostly pose larger loss values. To this end, we adopt the selection-based technique (Li et al., 2020) from noisy label learning to train a robust SLM for knowledge distillation.

Formally, after a few warm-up epochs with standard training on noisy labels, given the standard cross-entropy loss $l_i$ that reflects how well the model fits the sample $x_i$, we fit a two-component GMM to the loss $l_i$ to find out those clean samples. Let $w_i = p(g \mid l_i)$ represent the probability of $x_i$ belonging to the Gaussian component with smaller mean $g$, which can also be deemed as its clean probability. Then we divide the training dataset into a clean subset and a noisy subset by setting a threshold $\tau$ on $w_i$, which is considered as a labeled set $\mathcal{D}_l$ and a noisy set $\mathcal{D}_u$ respectively,

$$
\begin{aligned}
\mathcal{D}_l &= \{(x_i, \tilde{y}_i) \mid x_i \in \mathcal{D}_{\text{train}}, w_i \geq \tau\}, \\
\mathcal{D}_u &= \{(x_i) \mid x_i \in \mathcal{D}_{\text{train}}, w_i < \tau\}
\end{aligned} \tag{3}
$$

To improve the robustness of training, we utilize consistency regularization for boosted performance, which assumes that a classifier should produce a similar prediction on a local neighbor of each data point. Given an input $x_i$, we adopt back-translation (Sennrich et al., 2016) to paraphrase it and obtain the augmented version $x_i^{\text{aug}}$. For the labeled and unlabeled data, the consistency regularizations are formulated,

$$
\begin{aligned}
L_{\text{cr}}^l &= \frac{1}{|\mathcal{D}_l|} \sum_{x_i \in \mathcal{D}_l} l_{\text{ce}}(x_i^{\text{aug}}, \tilde{y}_i), \\
L_{\text{cr}}^u &= \frac{1}{|\mathcal{D}_u|} \sum_{x_i \in \mathcal{D}_u} l_{\text{kl}}(S(x_i^{\text{aug}}), S(x_i))
\end{aligned} \tag{4}
$$

where $l_{\text{ce}}$ and $l_{\text{kl}}$ are standard cross entropy and KL divergence respectively. Finally, the total loss for self-training of SLM is aggregated,

$$
L_{\text{total}} = L_{\text{clean}} + \alpha(L_{\text{cr}}^l + L_{\text{cr}}^u) \tag{5}
$$

where $L_{\text{clean}}$ is the cross entropy loss on $\mathcal{D}_l$, $\alpha$ is the loss weight parameter. We refer readers to Appendix B.1 for more implementation details.

### 4.2.2 Demonstration Pool Filtering

While the SLM $S$ can filter out a clean subset to enhance its performance during self-training, other stubborn noisy labels are hard to correct by SLM itself due to the confirmation bias. Thanks to our robust SLM, we can filter out those clean and representative samples and construct a high-quality demonstration pool $\mathcal{D}_{\text{demo}}$ for the LLM to refurbish its potentially wrong predictions in previous rounds. One may directly reuse the GMM-based selection criterion again and take $\mathcal{D}_l$ as demonstrations. However, such a selection procedure is too aggressive since excessively over-selecting some noisy samples may still improve the self-training procedure. To this end, we would like to filter out a more curated $\mathcal{D}_{\text{demo}}$ that prioritizes representative examples with accurate labels to be included.

The construction process mainly contains two steps in a class-wise manner to cover every class and ensure diversity. For the training subset $\mathcal{D}_{\text{train}}^j$ of class $j$, following the memory effect of DNNs (Zhang et al., 2017), we utilize the small loss criterion and select samples with the smallest cross-entropy loss $l_i$ in the first $R$ percent to construct $\mathcal{D}_{\text{clean}}^j = \{(x_i, \tilde{y}_i) \mid rank(l_i) \leq R\%, \tilde{y}_i = j\}$. In practice, we set a small $R$ to ensure the high precision of $\mathcal{D}_{\text{clean}}^j$. Secondly, we further adopt a simple clustering algorithm $k$-medoids on the embeddings of SLM to filter out the most representative medoids samples from $\mathcal{D}_{\text{clean}}^j$ to construct $\mathcal{D}_{\text{demo}}^j$. When the $k$-medoids algorithm gets converged, the medoids of $k$ clusters are collected as $\mathcal{D}_{\text{demo}}^j$. Finally the integral demonstration set is merged from each class as $\mathcal{D}_{\text{demo}} = \cup_{j \in Y} \mathcal{D}_{\text{demo}}^j$.

With a high quality $\mathcal{D}_{\text{demo}}$, the great potential of LLM $P$ can be unleashed to refine those noisy-prone samples $\mathcal{D}_{\text{noisy}} = \mathcal{D}_{\text{train}} \setminus (\cup_{j \in Y} \mathcal{D}_{\text{clean}}^j)$ via few-shot ICL as described in section 4.1.

## 5 Experiment

In this section, we provide the experimental results to verify the effectiveness of the FreeAL framework. More results, including visualizations and model selection, can be found in Appendix.

### 5.1 Setup

**Datasets.** We evaluate the performance of FreeAL on both sequence-level and token-level tasks. For sequence-level tasks, we choose SST-2 (Socher et al., 2013), MR (Pang and Lee, 2005) dataset for sentiment classification, SUBJ (Pang and Lee, 2004) dataset for subjectivity classification and TREC (Voorhees and Tice, 2000) for topic classification. For token-level tasks, CoNLL03 (Tjong Kim Sang and De Meulder, 2003) dataset

| Model | Round | Demons/Annos | SST-2 | MR | SUBJ | TREC | CoNLL03 | MA | BC5-C | BC5-D |
|-------|-------|--------------|-------|------|------|------|---------|------|-------|-------|
| GPT-3.5-Turbo | 0 | Zero-shot | 88.93 | 89.99 | 57.11 | 43.36 | 64.19 | 59.51 | 69.28 | 27.74 |
| | 1 | Self-generated | 92.16 | 91.74 | 86.54 | 70.74 | 70.89 | 59.78 | 81.05 | 47.12 |
| | 3 | Selected by Round 2 | **94.93** | **92.89** | **90.33** | **77.70** | **74.71** | **61.38** | **82.40** | **52.59** |
| RoBERTa | 2 | Annotated by Round 1 | 94.70 | 92.43 | 92.24 | 76.75 | 74.49 | 61.41 | 81.61 | 52.89 |
| | 4 | Annotated by Round 3 | **95.49** | **92.64** | **92.85** | **81.59** | **78.79** | **62.15** | **82.81** | **59.25** |

Table 1: Comparisons of **transductive performance on training datasets** of different tasks. BC5-C/D refers to BC5CDR-Chemical/Disease dataset. For the token-level NER tasks (including CoNLL03, BC5-C, BC5-D) the F1-score is given and For the other sequence-level tasks the test accuracy is provided.

| Model | Round | Demons/Annos | SST-2 | MR | SUBJ | TREC | CoNLL03 | MA | BC5-C | BC5-D |
|-------|-------|--------------|-------|------|------|------|---------|------|-------|-------|
| GPT-3.5-Turbo | 0 | Zero-shot | 92.47 | 90.05 | 55.65 | 77.20 | 66.47 | 59.71 | 67.85 | 29.60 |
| | 1 | Self-generated | 93.73 | 90.85 | 83.85 | 80.00 | 70.22 | 59.97 | 76.90 | 50.68 |
| | 3 | Selected by Round 2 | **95.91** | **93.10** | **90.27** | **79.80** | **70.80** | **60.93** | **80.77** | **52.70** |
| RoBERTa | 2 | Annotated by Round 1 | 94.29 | 89.35 | 92.95 | 86.80 | 71.82 | 61.91 | 80.55 | 53.38 |
| | 4 | Annotated by Round 3 | **94.66** | **90.20** | **94.45** | **91.40** | **76.12** | **62.64** | **81.13** | **58.90** |

Table 2: Comparisons of **inductive performance on test datasets** of different tasks. BC5-C/D refers to BC5CDR-Chemical/Disease dataset. For the token-level NER tasks (including CoNLL03, BC5-C, and BC5-D) the F1-score is given and For the other sequence-level tasks the test accuracy is provided.

is adopted for named entity recognition (NER). To validate the feasibility of FreeAL in practical scenarios such as medical diagnosis and biochemical applications that demand highly specialized domain-specific expertise, we also conduct experiments on BC5CDR (Li et al., 2016) dataset with chemical and disease interactions as token-level NER tasks and Medical Abstract (MA) (Schopf et al., 2022) dataset describing 5 different classes of patient conditions as sequence-level classification task. More details are listed in Table 4.

**Performance Evaluation.** In this work, we evaluate FreeAL from two aspects: (i)-**Transductive Performance:** Given unsupervised training data, we evaluate the training accuracy of FreeAL which reflects how well task-specific knowledge is distilled; (ii)-**Inductive Generalization:** utilize the derived models, including the SLM and $\mathcal{D}_{demo}$ for LLM, to further assess the generalization efficiency on the unseen test set with the inductive learning paradigm. We report the classification accuracy or the F1 score on both training and testing sets. We test the performance at different rounds. Round 0 denotes vanilla zero-shot learning of LLM. Round 1 and round 2 denote the performance of LLM and SLM in the first training loop, while round 3 and 4 are those of the second refinery training loop, as shown in Figure 2. For all experiments, we run three times and report the averaged results.

**Baselines.** We compare FreeAL with multiple zero-shot and supervised baselines for LLMs and SLMs respectively. For LLMs, they are vanilla zero-shot ICL without demonstrations (Brown et al., 2020), supervised ICL with standard demonstration retrieval (Liu et al., 2022) from human-labeled training data, and supervised ICL with k-medoids to first filter a representative subset for demonstration retrieval. For SLMs, they are zero-shot distillation (Hinton et al., 2015; Smith et al., 2022) that finetunes the SLMs by using the annotations from zero-shot ICL of LLM as ground-truths, and standard supervised fine-tuning that finetunes the SLM with human-labeled data. We also compare FreeAL with some traditional active learning baselines in section 5.3.1, including (1) Random: It acquires annotation of to-be-labeled data randomly. (2) Entropy (Holub et al., 2008): It is the most commonly used uncertainty-based baseline that acquires samples with the highest predictive entropy. (3) CAL (Margatina et al., 2021): It is a recent active learning method that acquires contrastive examples for pre-trained language models.

**Implementation Details.** We adopt OpenAI's GPT-3.5-Turbo language model, also known as ChatGPT, as our LLM and we use RoBERTa-Base from Huggingface Transformers (Wolf et al., 2020) as the downstream SLM. For the biomedical tasks including MA and BC5DER dataset, we utilize a

| Model | Ablation | Human | SST-2 | MR | SUBJ | TREC | CoNLL | MA | BC5-C | BC5-D |
|---|---|---|---|---|---|---|---|---|---|---|
| GPT-3.5-Turbo | Zero-shot ICL | ✗ | 92.47 | 90.05 | 55.65 | 77.20 | 66.47 | 59.71 | 67.85 | 29.60 |
| | **FreeAL (ours)** | ✗ | **95.91** | **93.10** | **90.27** | **79.80** | **70.80** | **60.93** | **80.77** | **52.70** |
| | Δ Absolute gain | - | +3.44 | +3.05 | +34.6 | +2.60 | +4.33 | +1.22 | +12.9 | +23.1 |
| | Supervised ICL (Standard) | ✓ | 96.06 | 92.85 | 89.30 | 81.50 | 85.46 | 61.22 | 82.24 | 68.63 |
| | Supervised ICL ($k$-medoids) | ✓ | 96.10 | 93.19 | 90.35 | 82.60 | 84.97 | 61.13 | 82.06 | 67.93 |
| RoBERTa | Zero-shot distillation | ✗ | 92.81 | 88.60 | 59.25 | 82.80 | 69.71 | 61.22 | 77.05 | 31.98 |
| | **FreeAL (ours)** | ✗ | **94.66** | **90.20** | **94.45** | **91.40** | **76.12** | **62.64** | **81.13** | **58.90** |
| | Δ Absolute gain | - | +1.85 | +1.60 | +35.2 | +8.60 | +6.41 | +1.42 | +4.08 | +26.9 |
| | Supervised FT | ✓ | 94.89 | 91.05 | 95.95 | 96.70 | 88.11 | 63.96 | 87.26 | 75.38 |

Table 3: Performance comparison of FreeAL with the zero-shot and supervised counterparts on the test dataset. BC5-C/D refers to BC5CDR-Chemical/Disease dataset. The results of FreeAL are in bold. Supervised FT refers to supervised fine-tuning. The absolute gain indicates the improvement of FreeAL compared to the zero-shot baseline.

| Dataset | Domain | #Token | #Train | #Test |
|---|---|---|---|---|
| SST-2 | Sentiment cls | 19.3 | 6,920 | 1,821 |
| MR | Sentiment cls | 21.6 | 8,662 | 2,000 |
| SUBJ | Subjectivity cls | 24.5 | 8,000 | 2,000 |
| TREC | Topic cls | 10.2 | 5,452 | 500 |
| CoNLL03 | NER | 14.59 | 14,041 | 3,453 |
| MA | Medical cls | 205.3 | 11,550 | 2,888 |
| BC5CDR | NER | 25.92 | 4,560 | 4,797 |

Table 4: A list of benchmarks used in the experiments. **#Train** and **#Test** indicate the size of the training and test dataset. **#Token** is the number of tokens on average for the corresponding training dataset.

BioMed-RoBERTa-base (Gururangan et al., 2020) that is pre-trained on the Semantic Scholar corpus as SLM for boosted performance. For fair comparisons, all the ICL processes of the LLM comprise $m = 10$ context examples as demonstrations except on MA dataset 5 is adopted due to the maximum context length 4,096 for GPT-3.5-Turbo. The collaborative training process is performed on the training dataset first and then the fine-tuned SLM and the $\mathcal{D}_{\text{demo}}$ for LLM are utilized to be evaluated on the test dataset. More details of the robust self-training are put in Appendix B.2.

## 5.2 Main Results

Table 1 and Table 2 display the results of FreeAL at different rounds in the collaborative training progress on the training and test dataset for transductive and inductive performance respectively. Table 3 reports the comparisons of FreeAL with other zero-shot and supervised counterparts.

Based on these results, it can be observed that *FreeAL significantly enhances the unsupervised performance of both LLM and SLM*. Free exceeds

the zero-shot ICL for the LLM by 3.44%, 3.05%, and 2.60% on the SST-2, MR, and TREC dataset respectively. It reaches a staggering lead of 34.6% on the SUBJ dataset where the LLM fails to adapt to on its own. In the medical diagnosis and biochemical fields, FreeAL also exhibits a notable advantage of 12.9% and 23.1% on the chemical and disease tasks of the BC5CDR dataset. FreeAL showcases a similar trend of leading performance for the SLMs. Interestingly, In comparison to the supervised counterparts, FreeAL achieves competitive performance on par with these supervised rivals on some simple tasks such as SST-2 and SUBJ datasets and greatly narrows the gap between the zero-shot and fully-supervised performances on other challenging tasks. Notably, the performance can be further improved with more interaction rounds (also larger cost), but 4 rounds of interaction can achieve satisfactory results empirically. These results suggest that FreeAL is able to fully distill the task-related knowledge from LLMs' weak supervision. More analyses can be found in Section 5.3.2.

## 5.3 Analysis

### 5.3.1 Comparisons with Active Learning

We also compare our FreeAL framework with some traditional active learning methods on the SST-2 and MR dataset. As shown in Table 5 and Figure 1, It can be observed that FreeAL outstrips the traditional active learning baselines with 20% and 50% acquired human annotations, which further indicates that FreeAL can serve as a superior alternative to traditional active learning by leveraging the rich knowledge of LLMs as a low-cost human-free supervision source.

| Method | Human Anno | SST-2 | MR |
|--------|:----------:|:-----:|:--:|
| Random | 20% samples | 92.42 | 88.10 |
|        | 50% samples | 92.92 | 89.10 |
| Entropy | 20% samples | 92.37 | 88.65 |
|         | 50% samples | 94.29 | 90.00 |
| CAL | 20% samples | 93.36 | 88.45 |
|     | 50% samples | 94.56 | 89.75 |
| **FreeAL (ours)** | ✗ | **94.66** | **90.20** |

Table 5: Comparisons of FreeAL with traditional active learning algorithms on the SST-2 and MR dataset.

### 5.3.2 Effect of Collaborative Training

From a more nuanced perspective of the performance improvements at different rounds on the training set in Table 1, it can be noticed that FreeAL iteratively refines the noisy annotations during the collaborative training. The improvement from round 0 to 1 indicates the effectiveness of self-generated demonstrations for better initial annotations. The performance advancements from rounds 1 to 2 and rounds 3 to 4 demonstrate the ability of robust self-training for SLM to distill valuable knowledge from noisy annotations. Further, the performance boost from round 2 to round 3 verifies the efficacy of the active label refinery process. For the test dataset in Table 2, the performance changes follow a similar trend with some fluctuations, which have been further discussed in Appendix A.1.

To further validate the efficacy of collaborative training, we also conduct additional ablation experiments for the components of FreeAL as shown in Table 6. For the generalization performance on the SLM, we compare FreeAL with its variant that discards robust self-training and adopts the standard cross entropy loss for training (including round 2 and 4). It can be observed that robust self-training largely improves the performance of FreeAL. For the performance of LLM, we ablate FreeAL with other selection strategies from traditional active learning rather than small loss selection, including random selection and entropy selection that selects samples with the lowest entropy values with the same budget as small loss selection. We can see that entropy selection slightly makes up for the poor performance of random selection, but still lags behind FreeAL by a notable margin.

### 5.3.3 Impact of In-Context Examples $m$

Then, we show the effect of different numbers $m$ of in-context examples during the process of ICL

| Model | Ablation | SST-2 | MR |
|-------|----------|:-----:|:--:|
| RoBERTa | **FreeAL** | **94.66** | **90.20** |
|         | w/o robust self-training | 89.18 | 88.95 |
| GPT-3.5-Turbo | **FreeAL** | **95.91** | **93.10** |
|               | with random selection | 95.12 | 92.15 |
|               | with entropy selection | 95.65 | 92.67 |

Table 6: Ablation study of FreeAL for the SLM and LLM on the SST-2 dataset and MR dataset.

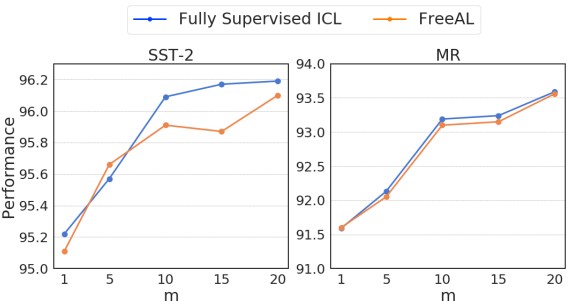

Figure 3: Ablation study of different numbers of in-context examples $m$ on the SST-2 and MR dataset.

on the SST-2 and MR datasets. As shown in Figure 3, FreeAL is able to produce a competitive performance to the supervised rivals over a wide range of $m$ from 1 to 20, this further verifies the robustness of FreeAL and we can simply adopt $m = 10$ for fair comparisons in our experiments.

## 6 Conclusion

In this work, we overhaul the traditional active learning in the era of LLMs and propose a novel framework called FreeAL that merely relies on the knowledge of LLMs to enhance human-free generalization performance. The key idea of FreeAL is to distill and filter the task-related knowledge from LLMs with a collaborative framework, where the LLM is employed as an active annotator and the SLM is engaged as a weak learner to filter out valuable samples for label refinery. The empirical results indicate that FreeAL can largely improve unsupervised performance and reaches comparable performance with supervised rivals in some tasks. While our FreeAL framework operates autonomously without human supervision, it is flexible and can be easily boosted with additional limited human supervision, which we leave for our future work. We hope that our work can spark heightened interest in developing new active annotation algorithms in the era of LLMs.

## Limitations

Our proposed FreeAL is a collaborative framework that aims to enhance unsupervised performance without human effort. Despite its effectiveness, there is still much potential for improvement. First, the effectiveness of FreeAL largely hinges on the strong ability of LLMs. For some domains that are extremely challenging or eccentric, the commonly adopted GPT-3.5-Turbo nowadays may fail to provide a qualified initial annotation, even with self-generated demonstrations. Our model is anticipated to be suitable for these circumstances with the advancement of more powerful LLMs across diverse domains. Besides, we thoroughly forgo human efforts in our FreeAL framework while in practical scenarios there may exist more or less available human support. It remains underexplored how to effectively combine the supervision from human experts and LLMs to synergize their individual strengths, and we leave it for our future work.

## Ethics Statement

While our proposed FreeAL serves as an innovative way to enhance generalization performance without human intervention, the predictions and self-generated demonstrations of the adopted LLM API may include bias and unfairness. Indeed, if one utilizes FreeAL with such biased annotations, it may unpleasantly yield unfair and biased predictions based on characteristics like race, gender, disabilities, LGBTQ, or political orientation. To alleviate this issue, we recommend that potential users first use bias reduction and correction techniques to remove biased text and predictions so as to improve overall fairness and ethical standard.

## Acknowledgements

This work is majorly supported by the NSFC under Grants (No. 62206247), and in part by the National Key Research and Development Program of China (No. 2022YFB3304101). Junbo Zhao also thanks the sponsorship by the Fundamental Research Funds for the Central Universities (No. 226-2022-00028). This paper is also supported by Netease Youling Crowdsourcing Platform[1]. As the importance of data continues rising, Netease Youling Crowdsourcing Platform is dedicated to utilizing various advanced algorithms to provide high-quality, low-noise labeled samples.

---

[1]https://fuxi.163.com

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

# A  Additional Experimental Results

## A.1  Discussion on Model Selection

With our collaborative training paradigm, we are able to interactively distill and filter task-related knowledge from LLMs. Empirically, our FreeAL method significantly enhances the zero-shot (distillation) performance of both SLMs and LLMs as discussed in Section 5. One intriguing finding is that, in the majority of evaluation cases, the final SLMs outperform the LLMs. This observation can be attributed to the superior distillation ability of SLMs during the weakly-supervised fine-tuning process. Consequently, we believe that SLMs remain a viable choice for practical deployment due to their impressive fine-tuned performance and low computational requirements. Furthermore, in more general scenarios, we recommend the utilization of a validation set to determine the most suitable model for deployment.

## A.2  FreeAL Can Reduce the Annotation Cost

As FreeAL solely depends on the knowledge of LLM and not on human efforts, it can naturally be leveraged as a low-cost data labeler in real-world scenarios. In Table 7, we evaluate the cost disparity between FreeAL and human annotations. Following previous work (Wang et al., 2021), for human labeling, it costs $0.11 per 50 input tokens with a minimum of $0.11. For FreeAL, the cost per example for $m$-shot inference is estimated approximately as $(\#Token \times (m+1) + 100) \times 2 \times (2 \times 10^{-6})$, where $\#Token$ is the average token numbers in Table 4, $(2 \times 10^{-6})$ is the cost for GPT-3.5-Turbo per token, 100 is roughly the tokens for the task-specific descriptions and the model reply. For each

| Source | SST-2 | MR | SUBJ | TREC | MA |
|--------|-------|-----|------|------|-----|
| Human | 0.11 | 0.11 | 0.11 | 0.11 | 0.55 |
| FreeAL | $1.2e^{-3}$ | $1.3e^{-3}$ | $1.5e^{-3}$ | $8.5e^{-4}$ | $4.5e^{-3}$ |

Table 7: Comparisons of annotation cost($) per example between human labeling and FreeAL.

| Model | Round | Annotations | SST-2 | MR |
|-------|-------|-------------|-------|-----|
| RoBERTa | - | **Vanilla FreeAL** | **94.66** | **90.20** |
| | 1 | Initial 10% from LLM | 87.97 | 81.20 |
| | 2 | Another 10% from LLM | 93.69 | 87.75 |
| | 3 | Another 10% from LLM | 93.76 | 88.95 |

Table 8: Results with multi-round annotation strategies.

sample, the ICL is performed at most twice as initial annotations and refined annotations. It can be observed that FreeAL can serve as a much cheaper data labeler while achieving passable performance.

When entrusted with a training set that is too large to label the entire dataset, **the annotation cost can be further reduced** by a simple multi-round solution. The core idea is to rely more on the weakly-supervised-learning capability of SLM to distill from a small number of annotated labels.

Specifically, for the initial annotation round of LLM, we randomly sample a subset of $P\%$ samples (empirically we set $P = 10$) to be annotated by LLM. After that, for robust self-training, we perform the original training process for the labeled data $D_{labeled}$ and simply extend the consistency regularization $L_{cr}^u$ for the noisy set $D_u$ to the originally unlabeled data (i.e., $D_u = D_u \cup D_{unlabeled}$). For the demonstration pool filtering, the construction process of $D_{demo}$ is the same, while for $D_{noisy}$ we randomly sample another subset of $P\%$ samples from the unlabeled samples to be annotated by LLM for the next iterations. The amount of iteration rounds can be larger than the original FreeAL if available to gradually distill the task-related knowledge with limited annotation cost.

As shown in the Table 8, such a simple remedy is able to achieve competitive results close to the original FreeAL with merely 10% of the previous cost each round, which proves the feasibility of FreeAL when we cannot afford to label the entire dataset. Notably, the process of randomly sampling the to-be-annotated subset on SLMs can be further improved with other advanced query strategies (e.g., uncertainty-based), which is a classic topic in traditional active learning.

| Model | Ablation | Round | SST-2 | MR |
|---|---|---|---|---|
| GPT-3.5-Turbo | **FreeAL** | Round 1 ⇒ 2 ⇒ 3 | 93.73 ⇒ **95.91** | 90.85 ⇒ **93.10** |
| | FreeAL w/o interaction | Round 1 ⇒ 3 | 93.73 ⇒ 95.37 | 90.85 ⇒ 92.15 |
| RoBERTa | **FreeAL** | Round 2 ⇒ 3 ⇒ 4 | 94.29 ⇒ **94.66** | 89.35 ⇒ **90.20** |
| | FreeAL w/o interaction | Round 2 ⇒ 4 | 94.29 ⇒ 94.27 | 89.35 ⇒ 89.55 |

Table 9: Ablation results of the interaction between the LLM and SLM for FreeAL on the SST-2 and MR dataset.

| Model | SST-2 | MR | SUBJ |
|---|---|---|---|
| RoBERTa-Base | 94.66 | 90.20 | 94.45 |
| RoBERTa-Large | 95.83 | 91.15 | 95.80 |

Table 10: Comparisons of FreeAL with different SLMs.

### A.3 Impact of SLM's Size

We also conduct experiments to reveal the impact of the size of SLM. As depicted in Table 10, when the size of SLM grows larger from RoBERTa-Base to RoBERTa-Large, FreeAL displays superior performance. This observation indicates that our FreeAL is compatible with different sizes of downstream SLM and the performance can be further improved with a larger SLM.

### A.4 Comparisons with Other AL Methods

Here we provide comparisons with some other active learning selection strategies, including Prob-Cover (Yehuda et al., 2022), BADGE (Ash et al., 2020), Region Entropy and Region CAL (Yu et al., 2022) in the Table 11. It can be observed that FreeAL exceeds all its rivals, which consistently demonstrates the superior performance of FreeAL.

### A.5 Comparisons with Dataset-generation-based Methods

We further supplement comparisons with some dataset-generation-based methods, including ZeroGen (Ye et al., 2022a), ProGen (Ye et al., 2022b) and SunGen (Gao et al., 2023). Our FreeAL is fundamentally different from them in several perspectives. First, these dataset-generation-based methods are tailored for an extreme scenario where training data is completely missing, which is unpractical in reality. Second, these methods typically generate low-quality samples, because they overlook the nuances and semantics present in the original authentic data. As a result, they mostly require generating a huge amount of synthetic data for decent performance. For example, on the SST-2 dataset, these methods generate 200k synthesized samples while authentic training samples are only

| Method | Human Anno | SST-2 | MR |
|---|---|---|---|
| ProbCover | 20% samples | 92.92 | 87.95 |
| | 50% samples | 93.49 | 89.75 |
| BADGE | 20% samples | 93.14 | 88.15 |
| | 50% samples | 93.97 | 89.90 |
| Region Entropy | 20% samples | 92.53 | 87.55 |
| | 50% samples | 94.03 | 88.75 |
| Region CAL | 20% samples | 92.37 | 88.20 |
| | 50% samples | 92.70 | 89.00 |
| **FreeAL (ours)** | ✗ | **94.66** | **90.20** |

Table 11: Comparisons of FreeAL with some other active learning algorithms on the SST-2 and MR dataset.

6.9k. Empirically, our FreeAL still outperforms these dataset-generation-based methods by a notable margin, as shown in Table 12.

### A.6 Results with More Distillation Methods

We also provide the comparisons with some other robust distillation methods, including GCE (Zhang and Sabuncu, 2018), SL (Wang et al., 2019) and ELR (Liu et al., 2020) in Table 13. We can see that FreeAL largely advances the performances of all these distillation baselines. Overall, FreeAL is designed as a flexible framework and we choose an empirically strong self-training algorithm for distillation to prove the feasibility of human-free active learning. One may design more power distillation algorithms for improved results, which we leave for future work.

### A.7 Effect of Interaction for LLM and SLM

To further demonstrate the importance of interaction between the LLM and the SLM. We provide the inductive performance for FreeAL without interaction. For the LLM, it directly adopts its own predictions on the training dataset at round 1 as the demonstration pool directly for testing. While the SLM employs its own predicted labels as supervision at round 2 directly for testing.

As displayed in Table 9, we observe that the SLM itself is hard to distill from its own predic-

| Method | SST-2 | SUBJ |
|---|---|---|
| ZeroGen | 87.27 | 80.45 |
| ProGen | 88.42 | - |
| SunGen | 89.45 | 83.25 |
| FreeAL (DitilBERT) | **91.82** | **92.15** |
| FreeAL (RoBERTa) | **94.66** | **94.45** |

Table 12: Comparisons of FreeAL with dataset-generation-based methods. We adopt DistilBERT as the SLM of FreeAL for fair comparisons.

| Method | SST-2 | MR |
|---|---|---|
| Zero-shot distillation | 92.81 | 88.60 |
| FreeAL with GCE | 93.68 | 88.90 |
| FreeAL with SL | 93.91 | 89.50 |
| FreeAL with ELR | 94.01 | 89.70 |
| **Vanilla FreeAL** | **94.66** | **90.20** |

Table 13: Comparisons with more distillation methods.

tions due to the inevitable confirmation bias, e.g., improves 0.2% compared with FreeAL's improvement of 0.85% on the MR dataset and even degrades on the SST-2 dataset. For the LLM, it can self-improve itself, but still underperforms our collaborative mechanism. Notably, LLM has an inescapable upper bound on the performance, according to our empirical findings where SLM outperforms LLM on 6 out of a total of 8 datasets. Such results indicate that the interaction between LLM and SLM can bring new opportunities to converge to a consensus result between them.

### A.8 Additional Visualization Results

We further provide some additional visualization results, including the transductive performance on the training dataset (i.e., the accuracy of pseudo labels) at different rounds in Figure 4 and the visualization of comparisons with traditional active learning methods on the MR dataset in Figure 5.

## B Additional Implementation Details

### B.1 More Details of Robust Self-training

During robust self-training, we also involve a mixup training strategy that interpolates the embeddings and the corresponding pseudo labels on the clean subset $\mathcal{D}_l$ to encourage linear behavior between samples. A virtual mixed training example is generated by linearly interpolating the randomly sampled pair of examples $(x_i, \tilde{y}_i)$ and $(x_j, \tilde{y}_j)$ in $\mathcal{D}_l$ and taking a convex combination of labels as the regression target,

$$\text{Emb}(x^m) = \sigma\text{Emb}(x_i) + (1-\sigma)\text{Emb}(x_j)$$
$$y^m = \sigma\tilde{y}_i + (1-\sigma)\tilde{y}_j \quad (6)$$

where $\text{Emb}(x_i)$ is the embedding of $x_i$ and $\sigma \sim \text{Beta}(\varsigma, \varsigma)$ and we simply set $\varsigma = 4$. The mixup loss is denoted as $L_{\text{mix}}$. the total loss for self-training of SLM is aggregated,

$$L_{\text{total}} = L_{\text{clean}} + \alpha(L_{\text{cr}} + L_{\text{mix}}) \quad (7)$$

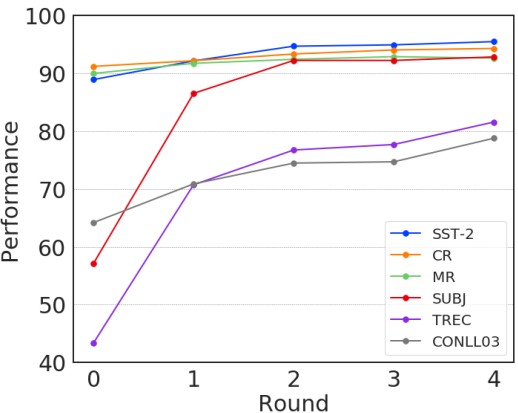

Figure 4: Performance of FreeAL on the training set at different rounds during collaborative training.

### B.2 More Implementation Details

In our experiments, for the LLM API, we follow the default official settings of the GPT-3.5-Turbo-0301 version. In the demonstration retrieval of ICL, we adopt the unsupervised embeddings with *bert-base-uncased* at the initial annotation round and the embeddings of SLM for later rounds. The construction and retrieval of $\mathcal{D}_{\text{demo}}$ are both performed in a class-wise manner to compose the final demonstrations. For the robust self-training of SLM, we adopt the hyperparameters either from previous works or fixed at a moderate value empirically without careful tuning. We finetune the SLM on the basis of the trainer of Huggingface for 50 epochs. The batch size is fixed at 32 with a maximum sequence length of 128. We adopt the AdamW optimizer with a learning rate selected from $\{3e-4, 3e-5, 3e-6\}$ and a weight decay of 0.01. For robust self-training, the threshold $\tau$ of GMM selection is fixed at 0.7 and the ratio $R$ of demonstration selection is fixed at 20. The loss weight parameter $\alpha$ is linearly ramped up from 0 to 1 to avoid overfitting false labels at the start. For evaluation of performance for LLM, as LLMs sometimes output ambiguous predictions outside the label space, these values are treated as random labels in the label space and repeated multiple times to evaluate the average performance

| Step | Prompt Details |
|---|---|
| **Demonstration Generation** | You are required to produce 100 English examples with labels for the task of text classification on the MR (Movie Review) dataset. These samples will be used as prompt examples for the GPT model. MR dataset is used in sentiment-analysis experiments and this dataset contains movie-review documents labeled with respect to their overall sentiment polarity (positive or negative). The task is to classify a movie review as positive or negative according to their overall sentiment polarity. For example, 100 of the unlabeled samples in MR dataset are as follows: ["review": "enigma is well-made , but it's just too dry and too placid ."] ["review": "the weakest of the four harry potter books has been transformed into the stronger of the two films by the thinnest of margins ."] ...... |
| **Active Annotation** | You are a helpful assistant for the task of text classification on the MR (Movie Review) dataset. You reply with brief, to-the-point answers with no elaboration as truthfully as possible. MR (Movie Review) dataset is used in sentiment-analysis experiments and this dataset contains movie-review documents labeled with respect to their overall sentiment polarity (positive or negative). Your task is to a binary classification to classify a movie review as positive or negative according to their overall sentiment polarity. The category is divided into two types: 'positive' and 'negative'. Given a movie review: <QUERY>. How do you feel about the sentiment polarity of the given movie review, is this positive or negative? please answer in a single line with 'positive' or 'negative'. |

Table 14: An example of prompt design on the MR dataset for the step of demonstration generation and active annotations. The in-context examples are omitted for the active annotation process here.

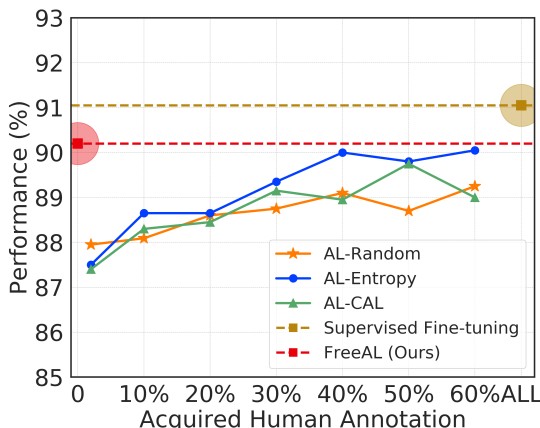

Figure 5: Comparisons of FreeAL with traditional active learning (AL) algorithms and supervised fine-tuning on the MR dataset. FreeAL surpasses all the active learning rivals and achieves near-supervised performance without human annotation.

selection process is performed on the token level in a different manner, we select those tokens with high confidence and matched predictions to pseudo-labels as clean and then filter out those samples whose tokens are all clean to constitute the clean subset. The consistency regularization and mixup loss are only suitable for the sequence-level tasks and are disabled in the token-level NER tasks.

### B.3 Prompt Design

We provide our prompt design on the MR dataset for the initial demonstration generation step and active annotation step in Table 14. Notably, we adopt the GPT-3.5-Turbo as our LLM so the prompts are also in the chat style with instructions.

during evaluation. Then in subsequent rounds, the SLMs adopt their own previous predictions to replace these ambiguous annotations of LLMs for robust self-training. For token-level tasks, as the