# OpenReview forum: "FreeAL: Towards Human-Free Active Learning in the Era of Large Language Models"
_EMNLP/2023/Conference — EMNLP 2023 Main_

### Official Review · Reviewer_3RFu · 2023-07-31

**Typos Grammar Style And Presentation Improvements:** 1. Line 20 in Algorithm 1 should be D…
**Soundness:** 4

**Excitement:**

4: Strong: This paper deepens the understanding of some phenomenon or lowers the barriers to an existing research direction.

**Missing References:**

I believe that it will be more comprehensive to include comparisons with zero-shot data generation methods (e.g. ZeroGen [3], ProGen [4] or SunGen [5]) in experiments. These methods cover a weaker setting (zero-shot), and can serve as better baselines to illustrate the novelty of FreeAL.

[3] Jiacheng Ye, Jiahui Gao, Qintong Li, Hang Xu, Jiangtao Feng, Zhiyong Wu, Tao Yu, and Lingpeng Kong. 2022. ZeroGen: Efficient zero-shot learning via dataset generation.

[4] Jiacheng Ye, Jiahui Gao, Zhiyong Wu, Jiangtao Feng, Tao Yu, and Lingpeng Kong. 2022. ProGen: Progressive zero-shot dataset generation via in-context feedback.

[5] Jiahui Gao, Renjie Pi, Yong Lin, Hang Xu, Jiacheng Ye, Zhiyong Wu, Weizhong Zhang, Xiaodan Liang, Zhenguo Li, Lingpeng Kong. 2022. Self-Guided Noise-Free Data Generation for Efficient Zero-Shot Learning.

**Paper Topic And Main Contributions:**

This paper proposes a collaborative learning framework, FreeAL, where an LLM serves as an active annotator to provide coarse-grained annotations, and a downstream SLM is trained over the pseudo labels and filter out high-quality samples to feedback LLM for in-context learning. It is notable that FreeAL runs in the unsupervised setting (only use unlabeled data), and can enhance both LLM’s and SLM’s performance.
Specifically, it first generates an initial demonstration set with LLM, and iterate in loop as (1) LLM annotates the whole unlabeled dataset, then (2) SLM is then trained with DivideMix technique [1], and used to filter out clean demonstration data with class-wise selection.

[1] Junnan Li, Richard Socher, and Steven C. H. Hoi. 2020. Dividemix: Learning with noisy labels as semi-supervised learning.

**Reasons To Accept:**

1. This paper provides a useful method for LLM-based unsupervised learning, which is practical in real-world scenarios. The authors also discuss the potential cost, compared to human annotations.
2. The idea to create an initial demonstration set with self-generation is interesting and effective.
3. The composition of several techniques (e.g. DivideMix and small loss selection) are well motivated and clearly ablated.

**Reasons To Reject:**

1. One of my main concerns is that FreeAL is incomparable with active learning methods, which only use a small portion of datasets for annotation efficiency and are blind on other data in unlabeled pool. Thus, I suggest to delete the comparison with active learning (Figure 1 and Table 5) or delay to Appendix, where discuss the label cost.
2. I further notice that FreeAL first annotates the whole dataset (D_{train}) and then reduce to the to-be-labeled set D_{noisy} iteratively. It is important to display the size of the to-be-labeled set to understand the cost of FreeAL at each round, since the API cost of ChatGPT is not negligible, especially given a large scale of training set.
3. It seems that the LLM’s boosted performance in FreeAL mainly attributes to better labels. However, as [2] suggests, even random labeled demonstration can yield similar LLM inference performance. I am not convinced that the boosted performance is reliable: it can be mainly an advantage of with text demo over without text demo (from round 0 to round 1) and authentic text (from unlabeled data) over synthetic text (from round 1 to round 3).
4. A suggestion, not necessarily weakness: have you experience with LLM/SLM self-improvement, without interaction? (1) I observe that round 2’s and round 3’s performance are almost the same, then can SLM still benefit from self-generated pseudo labels? (2) LLM can also self-determined “uncertain” samples, from token probability. It will be interesting to discuss more on this topic.

[2] Sewon Min, Xinxi Lyu, Ari Holtzman, Mikel Artetxe, Mike Lewis, Hannaneh Hajishirzi, and Luke Zettlemoyer. 2022. Rethinking the role of demonstrations: What makes in-context learning work?

**Reproducibility:**

4: Could mostly reproduce the results, but there may be some variation because of sample variance or minor variations in their interpretation of the protocol or method.

**Reviewer Confidence:**

4: Quite sure. I tried to check the important points carefully. It's unlikely, though conceivable, that I missed something that should affect my ratings.

---

> ### Author Rebuttal · Authors · 2023-08-29
>
> Thank you for your effort in reviewing our paper and providing constructive feedback. We are glad to see that you find the problem we study practical, our method well-motivated and effective. We provide detailed responses to address the concerns below.
>
> **Q1: Missing reference and comparisons**
>
> A1:  Thanks for the remark. We would like to clarify that these dataset-generation-based methods are fundamentally different from our FreeAL.
>
> * **Formulation and motivations:**  These dataset-generation-based methods [3-5] are tailored for an extreme scenario where training data is completely missing, which is unpractical in reality. In contrast, our FreeAL concentrates on how to generalize better to downstream tasks with the least human annotation given unlabeled authentic data. While data synthesizing is also encapsulated in FreeAL, it's a rather simple step to provide passable initial annotations for boosted performance, but we believe it doesn't affect our core contribution in providing an advanced active labeling framework.
> * **Data Quality:** We find that dataset-generation-based methods typically (and inevitably) generate low-quality samples, because they overlook the nuances and semantics present in the original authentic data, but only rely on the knowledge of LLM. This can largely hinder their generalization ability to specific downstream applications where the distributions deviate. To avoid this, these methods mostly assume the label distribution is known or uniform as a prerequisite, which is unrealistic in real-world deployment.
> * **Data Quantity:** Originates from the quality issue, these methods mostly require generating a huge amount of synthetic data for decent performance. For example, on the SST-2 dataset, these methods generate **200k synthesized samples** cumbersomely while the amount of authentic training samples is only 6.9k. This can result in a huge burden of computation and storage.
> * **Performance:** Despite a huge gap in the data size, our FreeAL still outperforms these dataset-generation-based methods by a notable margin. To see this, we compare our FreeAL with ZeroGen, ProGen and SunGen. We adopt DistilBERT as the SLM of FreeAL for fair comparisons. The experimental results on SST-2 and SUBJ are shown below, where we can see that FreeAL leads by a significant margin.
>
> | Method     &emsp; &emsp; &emsp; &emsp; &emsp;&emsp; &emsp;               | SST-2  &emsp; &emsp; &emsp; &emsp; &emsp;     | SUBJ    &emsp; &emsp; &emsp; &emsp; &emsp;         |
> | ----------------------- | --------- | -------------- |
> | ZeroGen                 | 87.27     | 80.45          |
> | ProGen                  | 88.42     | (Not provided) |
> | SunGen                  | 89.45     | 83.25          |
> | FreeAL (DistilBERT) | **91.82** | **92.15**      |
> | FreeAL (RoBERTa)   | **94.66** | **94.45**      |
>
> Therefore, we believe our work studies a rather different setup that can be much more practical and efficient with significantly better performance. Despite these advantages of FreeAL, we'd like to emphasize that our framework is designed to be a valuable plug-in helper that is flexible and compatible with different off-the-shelf data synthetic, distillation, and data selection algorithms. We'll add these new discussions and the missing reference to the revised version.
>
>
>
>
>
> **Q2: One of my main concerns is that FreeAL is incomparable with active learning methods, which only use a small portion of datasets for annotation efficiency and are blind on other data in unlabeled pool. Thus, I suggest to delete the comparison with active learning (Figure 1 and Table 5) or delay to Appendix, where discuss the label cost.**
>
> A2: Thanks for the remark. We kindly note that there might be some misunderstanding on the pipelines of traditional active learning that ```they actually also receives all training sample in every selection and training loop```. Specifically, a classical AL algorithm has the following steps in each loop:
>
> (1) Compute the metrics on **all (unlabeled) training samples**, and proactively select a subset it wants to learn from by ranking the metrics.
>
> (2) Label the subset by the external source --- human experts or oracle (cannot label the whole training set).
>
> (3) Training the SLMs with given data and labels, then loop to step (1).
>
> In summary, traditional AL is also allowed to access the whole training set like ours and the comparisons in Table 5 and Figure 1 are indeed fair. Notably, thanks to the great power and low cost of LLMs, our FreeAL can provide much more weak labels for SLMs than traditional AL. We believe it's our core superiority that can break the constraint of involving human labelers who can only annotate few samples.
>
>
>
> **Q3: I further notice that FreeAL first annotates the whole dataset (D_{train}) and then reduce to the to-be-labeled set D_{noisy} iteratively. It is important to display the size of the to-be-labeled set to understand the cost of FreeAL at each round, since the API cost of ChatGPT is not negligible, especially given a large scale of training set.**
>
> A3: Thanks for the question. The annotation cost of FreeAL is much lower than that of humans and can be further reduced.
>
> We first provide the to-be-labeled count for the LLM at the round 1 (i.e., the size of whole dataset $D_{train}$) and the round 3 (i.e., the size of $D_{noisy}$ filtered by GMM) in the table below. As also discussed in Appendix A.2, FreeAL can serve as a low-cost labeler which is much cheaper (requires about 1% cost) than human experts. When entrusted with a training set that is too large to label the entire dataset, the annotation cost can be further reduced by a simple multi-round solution where LLMs annotate a subset of samples in each loop. Please refer to `#Reviewer 3J9a.Q4` for more details.
>
> | Round  &emsp; &emsp; &emsp; &emsp;&emsp;| SST-2 &emsp; &emsp; &emsp; &emsp;&emsp; | MR  &emsp; &emsp; &emsp; &emsp;&emsp; |
> | ------- | ----- | ---- |
> | Round 1 | 6920  | 8662 |
> | Round 3 | 5537  | 6931 |
>
>
>
>
>
> **Q4：It seems that the LLM’s boosted performance in FreeAL mainly attributes to better labels. However, as [2] suggests, even random labeled demonstration can yield similar LLM inference performance. I am not convinced that the boosted performance is reliable: it can be mainly an advantage of with text demo over without text demo (from round 0 to round 1) and authentic text (from unlabeled data) over synthetic text (from round 1 to round 3).**
>
> A4: Thanks for the remark. We kindly disagree with the reviewer that we can use random labels to replace the true labels for any tasks. Consider a simple thought experiment:
>
> 1. Problem 1: We map the words 'A' to label 'Y1', 'B' to label 'Y2', 'C' to label 'Y3' ...
> 2. Problem 2: We map the words 'A' to label 'Y26', 'B' to label 'Y25' ...
>
> Can we provide random labels to solve any problem? Clearly, if one model achieves very high performance on Problem 1, its performance on Problem 2 can be extremely low. Therefore, random pairing texts and labels can potentially destroy the performance. We'd also like to note that the original paper [2] has concluded that the good performance of ICL possibly attributes to several aspects, including label space/distribution, language format, and also correctly-paired input-labels.
>
> From the empirical standpoint, we'd like to clarify that our better labels do indeed improve the ICL performance. To display it, we conduct experiments on the label enhancement process of LLM where the demonstrations are equipped with random labels. Specifically, we have stored and adopted the same self-generated or SLM-filtered demonstration pool $D_{demo}$ to ensure the comparisons are fair. As shown in the Table below, FreeAL enhances the performance with a much larger margin than FreeAL with random label, which demonstrates that the boosted performance is reliable and can be attributed to better labels. Importantly, randomly-labeled demonstrations can **even worsen the performance** compared to zero-shot ICL from round 0 to 1 on MR dataset and from round 1 to 3 on SUBJ dataset.
>
> | Dataset | Round  | Demonstrations           &emsp; &emsp; &emsp; &emsp; &emsp;&emsp; &emsp; &emsp;                        | FreeAL &emsp; &emsp; &emsp; &emsp;          | FreeAL with random label |
> | ------- | ----- | ------------------------------------------------------ | ------------------ | ------------------------ |
> | SST-2   | 0->1  | without $D_{demo}$ ->  with self-generated $D_{demo}$  | 92.47 -> **93.73** | 92.47 -> 93.22           |
> | SST-2   | 1->3  | with self-generated $D_{demo}$ -> authentic $D_{demo}$ | 93.73 -> **95.91** | 93.73 -> 94.89           |
> | MR      | 0->1  | without $D_{demo}$ -> with self-generated $D_{demo}$   | 90.05 -> **90.85** | 90.05 -> 89.30```(-0.75)```     |
> | MR      | 1->3  | with self-generated $D_{demo}$ -> authentic $D_{demo}$ | 90.85 -> **93.10** | 90.85 -> 91.95           |
> | SUBJ    | 0->1  | without $D_{demo}$ ->  with self-generated $D_{demo}$  | 55.65 -> **83.85**     | 55.65 -> 82.70           |
> | SUBJ    | 1->3  | with self-generated $D_{demo}$ -> authentic $D_{demo}$ | 83.85 -> **90.27**     | 83.85 -> 83.65```(-0.20)```       |
>
> Lastly, it should also be noted that the improvement of the LLMs is not our mere goal. In fact, through our extensive experiments, we found that the SLMs reach better performance on 6 of 8 benchmarks. This is attributed to our distillation step which trains the SLMs with better labels. Our work also proves that SLMs remain a viable choice for practical deployment due to their impressive fine-tuned performance and low computational requirements as discussed in Appendix A.1. Therefore, we believe better labels are still quite important in the collaborative distillation process of FreeAL. We'll add more results and discussions in our revised version.
>
>
>
> **Q5: A suggestion, not necessarily weakness: (1) have you experience with LLM/SLM self-improvement, without interaction?  I observe that round 2’s and round 3’s performance are almost the same, then can SLM still benefit from self-generated pseudo labels?**
>
> A5: That's a good suggestion, thank you! Below, we'd like to clarify the importance of interaction between the LLM and the SLM in our framework. The main reason is that deep networks are known to have the effect of confirmation bias, i.e., the mistakes in the pseudo-labels can accumulate and are nearly impossible to self-correct by the model itself. To alleviate this, we design a collaborative training framework with interaction to detect different labeling patterns by both LLMs (by activating their general knowledge) and SLMs (by self-training and distillation). Therefore, the interaction between LLMs and SLMs can bring new opportunities to converge to a consensus result between them.
>
> To reveal it more intuitively, we additionally provide the inductive performance without interaction in the table below. For the LLM, it directly adopts its own predictions on the training dataset at round 1 as the demonstration pool directly for testing. While the SLM employs its own predicted labels as supervision at round 2 directly for testing. We have two conclusions:
>
> 1. The SLM itself is hard to distill from its own predictions, e.g., improves 0.2 on the MR dataset compared with FreeAL’s improvement of 0.85. Even worse, SLM shows degraded performance when distilled from itself on the SST-2 dataset. It reveals that the SLM can be entangled in confirmation bias and fail to correct its own mistakes.
> 2. LLM can self-improve itself, but still underperforms our collaborative mechanism. e.g. FreeAL improves the self-distillation variant by 0.54 and 0.95 on the SST-2 and MR datasets. But, LLM has an inescapable upper bound on the performance, according to our empirical findings where SLM outperforms LLM on 6 out of a total of 8 datasets.
>
> | Model         | Ablation        | Round         | SST-2                | MR                   |
> | ------------- | --------------- | ------------- | -------------------- | -------------------- |
> | GPT-3.5-Turbo | FreeAL      | round 1->2->3 | 93.73 -> **95.91**   | 90.85 -> **93.10**   |
> | GPT-3.5-Turbo | FreeAL w/o interaction | round 1->3    | 93.73 -> 95.37       | 90.85 -> 92.15       |
> | RoBERTa       | FreeAL      | round 2->3->4 | 94.29 -> **94.66**   | 89.35 -> **90.20**   |
> | RoBERTa       | FreeAL w/o interaction | round 2->4    | 94.29 -> 94.27```(-0.02)``` | 89.35 -> 89.55 |
>
>
>
> **Q6: (2) LLM can also self-determined “uncertain” samples, from token probability. It will be interesting to discuss more on this topic.**
>
> A6: We originally aim to design a general framework which we have proved to be effective under the setting where the LLM is employed as black-box service model and token probability of generation is not provided, like ChatGPT and GPT-4. We thank the reviewer for providing such an interesting suggestion and we'll explore it in our future work.
>
>
>
> **Q7: In Table 1 and Table 2, the RoBERTa performances on CoNLL03 are exactly the same. Is this a mistake or coincidence?**
>
> A7: Thanks for pointing out it! The performances of RoBERTa on CoNLL03 in Table 2 should be 71.82 and 76.12 respectively at round 2 and round 4. We'll revise it in our revision as well as double-check our full manuscript.
>
>
>
> **Q8: I do not quite catch up with when eq.5 is used and when eq.7 is used. Can you explain a little more?**
>
> A8: Sorry for the confusion!  We originally aim to enhance the representation learning of SLMs so we adopt both consistency regularization and the mix-up augmentation in robust self-training as in Eq.(7). Actually, we have to note that the mix-up strategy is not our contribution, hence we omit it in Eq.(5) and delay it to the Appendix.
>
>
>
> **Q9: Typos: Line 20 in Algorithm 1 should be D_{demo} and D_{noisy}. Remove “-“ in line 391 and line 395.**
>
> A9: Our bad! Thanks for pointing them out. Line 20 in Algorithm 1 should be $D_{demo}$ and $D_{noisy}$. We will double-check and fix them in our revision!

---

### Official Review · Reviewer_3J9a · 2023-08-05

**Soundness:** 4

**Excitement:**

5: Transformative: This paper is likely to change its subfield or computational linguistics broadly. It should be considered for a best paper award. This paper changes the current understanding of some phenomenon, shows a widely held practice to be erroneous in someway, enables a promising direction of research for a (broad or narrow) topic, or creates an exciting new technique.

**Missing References:**

N/A

**Paper Topic And Main Contributions:**

This paper contributes FreeAL, an annotation-free active-learning-based approach for distilling knowledge from LLMs (GPT 3.5) into a student model (RoBERTa).
At a high level the approach consists of iteratitively repeating the following steps: 1. generating labels using in-context learning with an LLM, 2. training the student model on labels from the LLM, and 3. using the loss scores from the student model to partition the example set into "likely clean" labels (used in subsequent round as in-context examples) and "likely noisy" labels (which are relabeled by the LLM in subsequent rounds).
To completely mitigate the need for human annotations, the initial set of in-context examples is constructed using self-generated demonstrations from the LLM.
The paper provides experimental results on 8 classification datasets.
The first set of results illustrate how the FreeAL's performance improves as each of the above steps are carried out.
The second set of results compares FreeAL to a purely zero-shot setup and fully supervised baselines.
The third set of results compared FreeAL to active learning using human labeled examples.
The last set of results provides an ablation study measuring the benefit of the different design decisions, and number of in-context training examples.
Overall the results appear to be quite positive:
- In all cases FreeAL improves upon labeling/distilling using zero-shot learning, sometimes by quite a large margin (+20-30% absolute improvement in accuracy/f1)
- FreeAL is competitive with fully supervised performance on a number of tasks
- FreeAL outperforms active learning using human annotations
- The each step of the approach is shown to provide benefit.

**Questions For The Authors:**

- What is the superscript j in Algorithm 1?
- Related to my second reason to reject, I am somewhat curious about how FreeAL could be used in situations where you cannot afford to use an LLM to label the entire pool of data?

**Reasons To Accept:**

- Label generation/distillation using LLMs is currently recieving a lot of focus from the NLP community, and this paper provides an approach that significantly improves upon zero-shot learning without requiring any human annotations.
- The student model performance is often competitive with full finetuning.
- Given how complex the overall system is, I think the paper does a good job clearly explaining each component, and the experimentation (mostly) justifies each component's inclusion.

**Reasons To Reject:**

While I believe this is overall a strong paper, there are a few aspects of the experiments that negatively impact my evaluation of its soundness:
- "Robust self-training" involves noisy label learning as well as backtranslation (which I expect is a somewhat expensive procedure), however I did not find evidence supporting that both of these are necessary.
- I am not sure whether the comparisons presented in Table 5 are fair. If I understand correctly, FreeAL is allowed access to 100% of the data while the other methods only see a subset. I think it would be fairer if FreeAL were similarly limited.


**Reproducibility:**

4: Could mostly reproduce the results, but there may be some variation because of sample variance or minor variations in their interpretation of the protocol or method.

**Reviewer Confidence:**

4: Quite sure. I tried to check the important points carefully. It's unlikely, though conceivable, that I missed something that should affect my ratings.

**Typos Grammar Style And Presentation Improvements:**

Some of the language is a bit hyperbolic, e.g., saying the method "revolutionizes traditional active learning". I'd personally recommend toning it down.

- 218 - minutely
- 504 - tabel
- In tables 1 and 2, the numbers in parentheses look like references to equations rather than rounds.

---

> ### Author Rebuttal · Authors · 2023-08-29
>
> Thank you very much for your comments and suggestions! We are encouraged to see that you find our methods novel and effective. Below are our detailed responses.
>
> **Q1: "Robust self-training" involves noisy label learning as well as backtranslation (which I expect is a somewhat expensive procedure), however I did not find evidence supporting that both of these are necessary.**
>
> A1: Thanks for your remark. The noisy label learning and the back-translation (i.e., the consistency regularization) are widely adopted in the weakly supervised learning literature and are both components of the robust self-training for SLMs in our FreeAL, which has been shown to be effective in Table 6. We further conduct additional ablation experiments for these two components separately: 1) FreeAL w/o noisy label learning which forgoes the dataset division of GMM and regards all the samples as clean; 2) FreeAL w/o consistency regularization which discards the augmentation of backtranslation. It can be observed from the table below that the variants without either of these components suffer from perceptible performance degradation, but still beat the distillation and most of the active learning baselines by a notable margin.
>
> | Ablation                              | SST-2     | MR        |
> | ------------------------------------- | --------- | --------- |
> | FreeAL                          | **94.66** | **90.20** |
> | FreeAL w/o noisy label learning       | 93.97     | 89.45     |
> | FreeAL w/o consistency regularization | 93.81     | 89.25     |
> | Zero-shot distillation                | 92.81     | 88.60     |
>
> Notably, we originally tend to design a plug-and-play framework that is compatible with different distillation and augmentation strategies for SLMs, as discussed in the response to the ```#Reviewer 7Urc.Q2```. Hence we simply adopt the fashionable back-translation for augmentation, which is achieved directly via off-the-shelf translation models. Its cost can be further alleviated by adopting other augmentation methods instead (e.g., synonym replacement, random insertion/deletion). We'll add more results and discussions in our revised version.
>
>
>
>
>
> **Q2: I am not sure whether the comparisons presented in Table 5 are fair. If I understand correctly, FreeAL is allowed access to 100% of the data while the other methods only see a subset. I think it would be fairer if FreeAL were similarly limited.**
>
> A2: We kindly note that there might be some misunderstanding on the pipelines of traditional active learning that ```they actually also receives all training sample in every selection and training loop```. Specifically, a classical AL algorithm has the following steps in each loop:
>
> (1) Compute the metrics on **all (unlabeled) training samples**, and proactively select a subset it wants to learn from by ranking the metrics.
>
> (2) Label the subset by the external source --- human experts or oracle (cannot label the whole training set).
>
> (3) Training the SLMs with given data and labels, then loop to step (1).
>
> In summary, traditional AL is also allowed to access the whole training set like ours and the comparisons in Table 5 are indeed fair. Notably, thanks to the great power and low cost of LLMs, our FreeAL can provide much more weak labels for SLMs than traditional AL. We believe it's our core superiority that can break the constraint of involving human labelers who can only annotate few samples.
>
>
>
> **Q3:  What is the superscript j in Algorithm 1?**
>
> A3: Sorry for confusing the reviewer! The superscript j in Algorithm 1 refers to the class index as mentioned in line 343 of section 4.2.2. The construction process of the demonstration pool $D_{demo}$ is performed in a class-wise manner to ensure diversity, which filters out the demonstration sample class by class and then merges them to derive the demonstration pool. $\mathcal{D}_{\text{clean}}^j=\\{(x_i,\tilde{y}_i) \mid rank(l_i)\le R\\%,\tilde{y}_i=j \\}$ refers to the subset of clean samples with the observed labels equal $j$.
>
>
>
>
>
> **Q4:  Related to my second reason to reject, I am somewhat curious about how FreeAL could be used in situations where you cannot afford to use an LLM to label the entire pool of data?**
>
> A4:  Thanks for this remark! Firstly, we have to emphasize that the annotation cost of FreeAL is much lower than that the humans (about 1%)  on common labeling tasks, as discussed in Appendix A.2. This can be even more appealing when human annotation can be very expensive, e.g. medical analysis and industrial control.
>
> Secondly, we show this can also be reduced by a simple multi-round solution where LLMs annotate a subset of samples in each loop. The core idea is to rely more on the weakly-supervised-learning capability of the SLM to distill from a small number of annotated labels. Specifically, for the initial annotation round of LLM, we randomly sample a subset of P% samples (empirically we set P=10 here) from the training set to be annotated by LLM. After that, for robust self-training, we perform the original training process of FreeAL for the labeled data $D_{labeled}$ and simply extend the consistency regularization $L_{cr}^u$ for the noisy set $D_u$ to the originally unlabeled data (i.e., $D_u=D_u\cup D_{unlabeled}$). For the demonstration pool filtering, the construction process of $D_{demo}$ is the same, while for $D_{noisy}$ we randomly sample another subset of P% samples from the unlabeled samples to be annotated by LLM for the next iterations. The amount of iteration rounds can be larger than the original FreeAL if available to gradually distill the task-related knowledge with limited annotation cost.
>
> | Model       | Round | Annotations                          | SST-2     | MR        |
> | ----------- | :-----: | ------------------------------------ | --------- | --------- |
> | RoBERTa | - | Vanilla FreeAL                   | **94.66** | **90.20** |
> | RoBERTa     | 1     | Initial 10% samples annotated by LLM | 87.97     | 81.20     |
> | RoBERTa     | 2     | Another 10% samples annotated by LLM | 93.69     | 87.75     |
> | RoBERTa     | 3     | Another 10% samples annotated by LLM | 93.76     | 88.95     |
>
> As shown in the table above, such a simple remedy is able to achieve competitive results close to the original FreeAL with merely 10% of the previous cost each round, which proves the feasibility of FreeAL when we cannot afford to label the entire dataset. Notably, the process of randomly sampling the to-be-annotated subset on SLMs can be further improved with other advanced query strategies (e.g., uncertainty-based), which is a classic topic in traditional active learning.
>
>
>
> **Q5: Typos and presentation**
>
> A5: Thanks for your remark. We will double-check the typos and revise the inappropriate expressions in our revision to make them more self-contained.

---

### Official Review · Reviewer_7Urc · 2023-08-11

**Soundness:** 4

**Excitement:**

3: Ambivalent: It has merits (e.g., it reports state-of-the-art results, the idea is nice), but there are key weaknesses (e.g., it describes incremental work), and it can significantly benefit from another round of revision. However, I won't object to accepting it if my co-reviewers champion it.

**Paper Topic And Main Contributions:**

This paper proposes FreeAL, a human-free active learning framework that leverages powerful LLMs as data annotators. Specifically, FreeAL consists of active annotation, robust self-training, and filtering noisy data. Experimental results demonstrate the proposed FreeAL outperforms baseline methods with large margins.

**Reasons To Accept:**

1. This paper is well-written and well-organized.
2. The evaluation is comprehensive. Experimental results outperform baseline methods with large margins.
3. The idea is rather simple and straightforward.

**Reasons To Reject:**

1. Lacking comparisons with other baselines. There are many other AL sample-selection strategies, such as [1].
2. Leveraging LLMs as training data generators has been explored before. For instance, SunGen[2] and ZeroGen[3]. Comparison with these sota methods is necessary in my view. Also, the authors should consider some distillation methods as baselines as utilizing LLMs as annotators are kind of distillation.

[1] Yehuda, Ofer, et al. "Active learning through a covering lens." Advances in Neural Information Processing Systems 35 (2022): 22354-22367.

[2] Ye, Jiacheng, et al. "Zerogen: Efficient zero-shot learning via dataset generation." arXiv preprint arXiv:2202.07922 (2022).

[3] Gao, Jiahui, et al. "Self-Guided Noise-Free Data Generation for Efficient Zero-Shot Learning." The Eleventh International Conference on Learning Representations. 2022.

**Reproducibility:**

3: Could reproduce the results with some difficulty. The settings of parameters are underspecified or subjectively determined; the training/evaluation data are not widely available.

**Reviewer Confidence:**

3: Pretty sure, but there's a chance I missed something. Although I have a good feel for this area in general, I did not carefully check the paper's details, e.g., the math, experimental design, or novelty.

---

> ### Author Rebuttal · Authors · 2023-08-29
>
> Thanks very much for your time and effort in reviewing our paper and providing thoughtful feedback to improve our paper. We hope the following responses can clarify your confusion, and we are happy to provide additional explanations if needed.
>
> **Q1: Lacking comparisons with other baselines. There are many other AL sample-selection strategies, such as [1].**
>
> A1: Thanks for your advice. We have followed your advice to provide comparisons with some other AL sample-selection strategies, including ProbCover [1], BADGE [C1], Region Entropy and Region CAL [C2]. The results are shown in the table below. It can be observed that FreeAL exceeds all its rivals **despite they require 20% and 50% human-annotated data**, which consistently demonstrates the superior performance of FreeAL.
>
> | Method     &emsp; &emsp; &emsp;      | Human Annotation &emsp; &emsp;   | SST-2  &emsp; &emsp;     | MR   &emsp; &emsp;       |
> | -------------- | ---------------- | --------- | --------- |
> | FreeAL     | None         | **94.66** | **90.20** |
> | ProbCover      | 20% samples      | 92.92     | 87.95     |
> | ProbCover      | 50% samples      | 93.49     | 89.75     |
> | BADGE          | 20% samples      | 93.14     | 88.15     |
> | BADGE          | 50% samples      | 93.97     | 89.90     |
> | Region Entropy | 20% samples      | 92.53     | 87.55     |
> | Region Entropy | 50% samples      | 94.03     | 88.75     |
> | Region CAL     | 20% samples      | 92.37     | 88.20     |
> | Region CAL     | 50% samples      | 92.70     | 89.00     |
>
> *[C1] Ash J T, Zhang C, Krishnamurthy A, et al. Deep batch active learning by diverse, uncertain gradient lower bounds[J]. arXiv preprint arXiv:1906.03671, 2019.*
>
> *[C2] Yu Y, Kong L, Zhang J, et al. AcTune: Uncertainty-based active self-training for active fine-tuning of pretrained language models[C]//Proceedings of the 2022 Conference of the North American Chapter of the Association for Computational Linguistics: Human Language Technologies. 2022: 1422-1436.*
>
>
>
> **Q2: Leveraging LLMs as training data generators has been explored before. For instance, SunGen[2] and ZeroGen[3]. Comparison with these sota methods is necessary in my view.**
>
> A2:   Thanks for the remark. First, we'd like to clarify that these dataset-generation-based methods are fundamentally different from our FreeAL.
>
> - **Formulation and motivations:** These dataset-generation-based methods [2,3] are tailored for an extreme scenario where training data is completely missing, which is unpractical in reality. In contrast, our FreeAL concentrates on how to generalize better to downstream tasks with the least human annotation given unlabeled authentic data. While data synthesizing is also encapsulated in FreeAL, it's a rather simple step for improved performance, but we believe it doesn't affect our core contribution in providing an advanced active labeling framework.
> - **Data Quality:** We find that dataset-generation-based methods typically (and inevitably) generate low-quality samples, because they overlook the nuances and semantics present in the original authentic data, but only rely on the knowledge of LLM. This can largely hinder their generalization ability to specific downstream applications where the distributions deviate. To avoid this, these methods mostly assume the label distribution is known or uniform as a prerequisite, which is unrealistic in real-world deployment.
> - **Data Quantity:** Originates from the quality issue, these methods mostly require generating a huge amount of synthetic data for decent performance. For example, on the SST-2 dataset, these methods generate **200k synthesized samples** cumbersomely while the amount of authentic training samples is only 6.9k. This can result in a huge burden of computation and storage.
> - **Performance:** Despite a huge gap in the data size, our FreeAL still outperforms these dataset-generation-based methods by a notable margin. To see this, we compare our FreeAL with ZeroGen and SunGen. We adopt DistilBERT as the SLM of FreeAL for fair comparisons. The experimental results on SST-2 and SUBJ are shown below, where we can see that FreeAL leads by a significant margin.
>
> | Method  &emsp; &emsp; &emsp; &emsp; &emsp;              | SST-2 &emsp; &emsp; &emsp;        | SUBJ   &emsp; &emsp; &emsp;   |
> | ----------------------- | --------- | --------- |
> | ZeroGen                 | 87.27     | 80.45     |
> | SunGen                  | 89.45     | 83.25     |
> | FreeAL (DistilBERT) | **91.82** | **92.15** |
> | FreeAL (RoBERTa)    | **94.66** | **94.45** |
>
> Therefore, we believe our work studies a rather different setup that can be much more practical and efficient with significantly better performance. Despite these advantages of FreeAL, we'd like to emphasize that our framework is designed to be a valuable plug-in helper that is flexible and compatible with different off-the-shelf data synthetic, distillation, and data selection algorithms. We'll add our new empirical results and discussions to the revision.
>
>
>
>
>
>
>
> **Q3: Also, the authors should consider some distillation methods as baselines as utilizing LLMs as annotators are kind of distillation.**
>
> A3: For the distillation baselines, since we regard the LLM as a black-box model like GPT-3.5/4 where only the output text is accessible, we have compared our FreeAL with the zero-shot distillation baseline that finetunes the SLM with standard cross-entropy loss using the annotations from the ICL of LLMs as ground-truths in Table 3 of our original submitted version. We further provide the comparisons with some robust distillation methods, including GCE [C1], SL [C2] and ELR [C3] in the table below. We can see that FreeAL largely advances the performances of all these distillation baselines.
>
> | Method     &emsp; &emsp; &emsp; &emsp; &emsp;             | SST-2    &emsp; &emsp; &emsp;    | MR    &emsp; &emsp; &emsp;   |
> | ---------------------- | --------- | --------- |
> | Zero-shot distillation | 92.81     | 88.60     |
> | FreeAL             | **94.66** | **90.20** |
> | FreeAL with GCE        | 93.68     | 88.90     |
> | FreeAL with SL         | 93.91     | 89.50     |
> | FreeAL with ELR        | 94.01     | 89.70     |
>
> Despite our new results within a short period of response time, we'd like to clarify that the primary goal of our work is to investigate the feasibility of human-free AL in the era of LLMs. Thus, we designed our FreeAL to be a flexible framework. Specifically, we chose an empirically strong self-training algorithm for distillation that proves the feasibility of human-free AL. One may design more power distillation algorithms for improved results, which we leave for future work.
>
>
>
>
> *[C1] Zhang Z, Sabuncu M. Generalized cross entropy loss for training deep neural networks with noisy labels[J]. Advances in neural information processing systems, 2018, 31.*
>
> *[C2] Wang Y, Ma X, Chen Z, et al. Symmetric cross entropy for robust learning with noisy labels[C]//Proceedings of the IEEE/CVF international conference on computer vision. 2019: 322-330.*
>
> *[C3] Liu S, Niles-Weed J, Razavian N, et al. Early-learning regularization prevents memorization of noisy labels[J]. Advances in neural information processing systems, 2020, 33: 20331-20342.*

---

### Meta-Review · Area_Chair_oQJw · 2023-09-19

**Recommendation:** 5

**Metareview:**

This paper presents an active learning based framework FreeAL to distill knowledge from LLM to SLM where LLM serves as an active annotator and SLM as the student model. The proposed framework eliminates the need for human-labeled supervision. There is a unanimous consensus among reviewers that this is a sound approach for knowledge distillation without human annotations and the experiment result is very solid and positive. The rebuttal has effectively addressed reviewers' concerns by adding more analysis and baseline comparisons. We encourage the authors to take the feedback to revise the final version.

---

### Decision · Program_Chairs · 2023-10-07

**Decision:**

Accept-Main

**Comment:**

This paper presents an active learning based framework FreeAL to distill knowledge from LLM to SLM where LLM serves as an active annotator and SLM as the student model. The proposed framework eliminates the need for human-labeled supervision. There is a unanimous consensus among reviewers that this is a sound approach for knowledge distillation without human annotations and the experiment result is very solid and positive. The rebuttal has effectively addressed reviewers' concerns by adding more analysis and baseline comparisons. We encourage the authors to take the feedback to revise the final version.